# DM3D: Parameter-Efficient and Lossless 3D Object Detection via Distortion Minimization

## Abstract

Recent advancements in 3D deep learning have garnered significant attention, given their superior performance in fields like AR/VR, autonomous driving, and robotics. However, as the models and point cloud data continues to scale up, managing computational and memory demands becomes a critical challenge, particularly for real-world applications with strict latency and energy requirements. Previous methods have primarily focused on reducing computational costs and memory usage by addressing spatial redundancy, *i.e.*, filtering out irrelevant points or voxels. In contrast, this work presents a novel post-training weight pruning technique tailored specifically for 3D object detection. Our approach stands out in two key ways: (1) it operates independently from existing point cloud sparsification methods, targeting redundant parameters in pre-trained models that minimally affect both spatial accuracy and detection confidence (collectively referred to as "detection distortion"), and (2) it provides a flexible, plug-and-play framework compatible with other sparsity schemes including spatial sparsity and with any 3D detection model. Our method reduces detection distortion by employing a second-order Taylor approximation to identify layer-wise sparsity, allowing for a substantial reduction in model complexity without sacrificing detection accuracy. To efficiently manage the necessary second-order information, we devised a lightweight algorithm to gather Hessian information, followed by dynamic programming to optimize layer-wise sparsity allocation. Extensive experiments on the KITTI, nuScenes, and ONCE datasets validate the effectiveness of our approach, where we not only preserve detection performance but also notice enhancement while significantly reducing computational overhead. Noticeably, we achieve FLOPs reductions for Centerpoint model of as much as $3.89\times$ and $3.01\times$ on ONCE and nuScenes datasets respectively, without noticeable loss in mean Average Precision (mAP), and at most $1.65\times$ reduction **losslessly** for PVRCNN model on the ONCE dataset, thus pushing the boundaries of state-of-the-art performance.

## 1 Introduction

3D deep learning has gained great interest from both research and industry for its wide applications in autonomous driving, robotics and VR/AR, etc. Particularly, 3D object detection is one of the essential visual tasks for autonomous driving systems to understand the driving environment, which serves as the foundation for the subsequent decision-making process. Recent advances in LiDAR-based object detection Yan et al. (2018); Lang et al. (2019); Shi et al. (2020); Ye et al. (2020); Yi et al. (2020); Ao et al. (2021); Yin et al. (2021); Mao et al. (2021); Yang et al. (2020); Chen et al. (2023b) show the demand of significantly costly computation to meet an empirically acceptable accuracy, particularly in handling the challenges posed by highly heterogeneous and unstructured 3D data. However, to maximally mitigate hazardous events in real-world driving, fast inference is crucial for achieving low-latency detection. Therefore, accelerating the object detection on 3D data with high computation and memory demand to make them more feasible in real applications becomes an urgent task.

Past attempts Contributors (2022); Tang et al. (2023) exploited spatial sparsity in 3D point cloud modality and skipped unnecessary computation to obtain acceleration. Other attempts further identified unimportant points and voxels from raw LiDAR data to cut down memory footprint Sun et al.

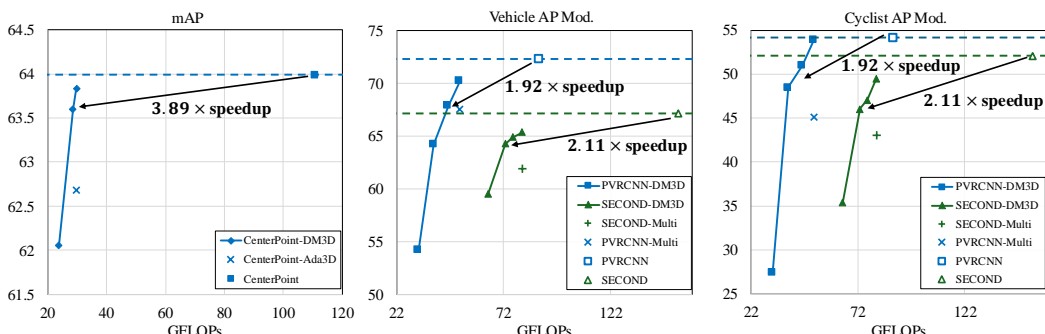

Figure 1: AP scores v.s. FLOPs on ONCE dataset of our methods DM3D compared to baselines. Dash lines are the dense model performances. The speedup measured by FLOPs reduction from dense models.

(2021); Liu et al. (2022a); Zhao et al. (2023). All of such exploitation only considers removing computation in voxel-wise connections, agnostic to explicit detection precision preservation in their formulations. The latest state-of-the-art sparse 3D object detection method Zhao et al. (2023) set a record of 20% floating-point operations per second (FLOPs) reduction achieved via removing spatial redundancy. However, there is an inevitable upper bound regarding to the FLOPs reductions when only cutting down point cloud information. On the other hand, the redundancy in the weights of 3D models remains notably high and has not been adequately addressed. Existing weight sparsification method for *3D segmentation task* He et al. (2022) pruned out convolution kernel connections with least neighbouring point access rate. However, such hit-rate-based weight selection scheme is sub-optimal on maintaining accuracy compared to typical magnitude-based and Taylor-based ranking schemes Lee et al. (2018; 2020); Xu et al. (2023) that have been proven to achieve up to over 90% on benchmark datasets like CIFAR-10. Another challenge in weight sparsification for 3D models is how to allocate layer-wise pruning ratios such that minimizes the negative impact on model accuracy. He et al. (2022) addressed this problem by employing a greedy search on single-layer rate-distortion relations. Although they have relaxed the allocation problem into sub-problems of each consecutive two layers groups to make the solution tractable, such relaxation still requires extensive real data collections, undermining the efficiency.

In this paper, we propose a generalized weight pruning framework for 3D object detection models. Inspired by classic rate-distortion theory, our approach minimizes the detection distortion of 3D detection models, including the detection locality and confidence, under the constraint of computation complexity (FLOPs). The pruning of weights is performed in a layer-wise manner where the pruning ratio of each layer is decided according to a Hessian-based rate-distortion score, which can be calculated efficiently by pre-computed gradient values. We then develop an ultra-fast dynamic programming algorithm with polynomial time complexity to find the globally optimal pruning ratio of each layer. Note that our proposed weight pruning and the previously proposed input pruning approaches can compensate with each other. By incorporating both weight and input pruning, we expect to obtain a maximal compression ratio. We compare with previous arts and perform extensive experiments on various 3D detection models to demonstrate the effectiveness of our approach. As shown in Fig. 1, we achieve remarkable speedups with negligible performance drops or even gains on various datasets. To the best of our knowledge, this is the first work that systematically proposes a weight pruning approach for 3D detection models in a distortion-minimized manner. We summarize the main contribution of our paper as follows:

- We propose a generic weight pruning framework for 3D object detection models capable of reducing computation complexity (FLOPs) via sparsifying weights. The pruning objective is formulated as a Pareto-optimization, which explicitly minimizes distortions of both detection bounding box locality and classification confidence. The proposed pruning scheme can be applied as a standalone plug-and-play module for post-train processing for the 3D objection model, and can also serve complementary with other spatial pruning methods.
- Our approach adopts a hessian-based layer-wise pruning scheme. Through performing dynamic programming and fine-grained optimization, we derive an extremely efficient algorithm with polynomial time complexity to find the global optimal solution. Thanks to the

distortion-minimized objective and the fine-grained optimization, our approach can maximally preserve detection accuracy under very high pruning ratio.

- Our approach achieves state-of-the-art performance on six 3D object detection models across three benchmark datasets.

## 2 RELATED WORKS

Typically, neural networks for 3D point clouds primarily employ either Point-based Qi et al. (2017) or Voxel-based approaches. Point-based methods directly process the raw 3D point cloud data using neural networks, whereas Voxel-based techniques first voxelize the raw data and then operate on the voxels rather than the original data. Numerous works Rukhovich et al. (2022); Yan et al. (2018); Shi et al. (2020); Deng et al. (2021); Yin et al. (2021); Chen et al. (2023b) took voxel-based approach for 3D object detection task.

### 2.1 SPATIAL SPARSITY

**Sparse 3D Convolution.** Sparse 3D Convolutional Neural Networks (CNNs) are designed to efficiently process sparse point cloud data, where a significant portion are background points with zero values. Therefore computing convolution on such data can be done by skipping empty positions in 3D space. Engelcke et al. (2017); Riegler et al. (2017) built hierarchical structures for efficient point cloud data representation. Regular Sparse Convolution (RSC) Graham (2014) and Submanifold Sparse Convolution (SSC) Graham & Van der Maaten (2017) were proposed to mitigate the increase of active sites in order to maintain spatial sparsity across Sparse Conv layers. Some libraries Tang et al. (2022); Yan et al. (2018) have been developed to support the Sparse 3D convolution for fast inference. Xu et al. (2020) applied different filters based on the input image's spatial location, addressing the inefficiencies of using standard convolutions on LiDAR data with spatially varying distributions.

**Spatial Redundancy Reduction.** It is easy to observe that natural redundancy in points/voxels still exist even in active regions of 3D data. Thereby, this approach aims to further identify redundant spatial data points to reduce computational complexity. Graham (2014) was an early work proposed that output points should be omitted when corresponding input points are absent, ensuring computations are only performed for areas with relevant input data. Guo et al. (2022) simplified set abstraction procedure for point-based 3D action recognition models to identify important points and frames. Chen et al. (2022); Liu et al. (2022a) proposed to engage feature sparsity by predicting importance maps to remove redundant regions, achieving computation and memory reductions. Multi Li et al. (2023) proposed a focal loss to predict voxel importance combined with voxel distillation. Ada3D Zhao et al. (2023) further leveraged the spatial redundancy in 2D BEV Backbone by proposing Sparsity Preserving BatchNorm to perform 3D-to-BEV feature transform. Liu et al. (2022b); Chen et al. (2023a) exploited the fact that the differences (residuals) between consecutive 3D frames are typically sparse.

### 2.2 SPARSITY IN MODEL WEIGHTS

Apart from exploiting the intuitive spatial redundancy, much less attempts has been done to leverage weight redundancy in 3D networks. Zhao et al. (2021) used a generator to propose pruning strategies and an evaluator that employs Bayesian optimization to select out pruning strategies for 3D object detection. They configured unique pruning schemes and rates for each layer. Not All NeighboursHe et al. (2022) identified kernel neurons that are least frequently attended to compute output features in 3D segmentation models. They also performed a layerwise sparsity allocation by greedy search on two-layer grouping strategy. $CP^3$ Huang et al. (2023) leveraged structured pruning (channel-wise pruning) for point-based 3D networks. None of existing weight pruning approaches is established on minimizing the impact of pruning on the detection performance as we proposed in this paper.

## 3 PRELIMINARIES

**Pruning within layer.** We target at pruning learnable parameters with unstructured sparsity for all feature extraction layers in 3D detection models, *i.e.*, the kernels in SparseConv Contributors

(2022) in the 3D backbone and Conv2d in the 2D Bird's Eye View (BEV) backbone. To determine which parameter in a layer needs to be pruned, given a layer sparsity, we rank the neuron within each layer by the absolute first-order Taylor expansion term and eliminate the bottom ranked ones. Mathematically, we first derive the neuron ranking score matrix by the Taylor expansion $\boldsymbol{S} = |\boldsymbol{W} \cdot \nabla_{\boldsymbol{W}} \boldsymbol{y}|$ Molchanov et al. (2019), where $\boldsymbol{W}$ denotes the network parameters and $\nabla_{\boldsymbol{W}} \boldsymbol{y}$ stands for the derivative of the network output $\boldsymbol{y}$ to the network parameters $\boldsymbol{W}$. The above pruning scheme can be formulated as $\widetilde{\boldsymbol{W}} = \boldsymbol{W} \odot \boldsymbol{M}_\alpha(\boldsymbol{S})$, where $\boldsymbol{M}_\alpha(\boldsymbol{S})$ is the binary mask generated from the ranking score matrix $\boldsymbol{S}$ under the pruning ratio $\alpha$. We further defines the weight perturbation $\Delta \boldsymbol{W} = \widetilde{\boldsymbol{W}} - \boldsymbol{W}$ caused by a typical pruning operation to the weight. We further adopt a basic assumption for the weight perturbation $\Delta \boldsymbol{W} = \widetilde{\boldsymbol{W}} - \boldsymbol{W}$ caused by a typical pruning operation to the weight:

**Assumption 1** *For the network with $l$ layers and the $i$-th layer with weight $\boldsymbol{W}^{(i)}$, **i.i.d. weight perturbation across layers** Zhou et al. (2018) which means the joint distribution across different layers is zero-meaned:*

$$\forall\, 0 < i \neq j < l,\, E(\Delta \boldsymbol{W}^{(i)} \Delta \boldsymbol{W}^{(j)}) = E(\Delta \boldsymbol{W}^{(i)}) E(\Delta \boldsymbol{W}^{(j)}) = 0, \tag{1}$$

*and also zero co-variance:* $E(\|\Delta \boldsymbol{W}^{(i)} \Delta \boldsymbol{W}^{(j)}\|^2) = 0$.

## 4 METHODOLOGIES

### 4.1 PROBLEM DEFINITION

Given a 3D detection model consisting of (1) a feature extractor $f$ of $l$ layers with the parameter set $\boldsymbol{W}^{(1:l)} = (\boldsymbol{W}^{(1)}, ..., \boldsymbol{W}^{(l)})$, where $\boldsymbol{W}^{(i)}$ is the weights in layer $i$ and (2) a detection head on top of the 2D/3D feature $f(x; \boldsymbol{W}^{(1:l)})$ that output 3D detection bounding box predictions $\boldsymbol{p}_b(f(x; \boldsymbol{W}^{(1:l)})) \in \mathbb{R}^{N_s \times S}$ and confidences $\boldsymbol{p}_c(f(x; \boldsymbol{W}^{(1:l)})) \in \mathbb{R}^{N_s \times C}$ where $N_s$ denotes the number of predicted bounding boxes, $C$ stands for the number of classes and $S$ represents the dimension of the bounding box coordinates. Therefore the detection output is the concatenation of both bounding box prediction and class confidence scores:

$$\boldsymbol{y} = [\boldsymbol{p}_b(f(x; \boldsymbol{W}^{(1:l)}))^\top, \boldsymbol{p}_c(f(x; \boldsymbol{W}^{(1:l)}))^\top]^\top. \tag{2}$$

Pruning parameters in the $f$ will give a new parameter set $\widetilde{\boldsymbol{W}}^{(1:l)}$. We view the impact of pruning as the distortion between the dense prediction $\boldsymbol{y}$ and the prediction $\tilde{\boldsymbol{y}}$ of the pruned model.

$$\boldsymbol{y} - \tilde{\boldsymbol{y}} = \begin{bmatrix} \boldsymbol{p}_b(f(x; \boldsymbol{W}^{(1:l)})) - \boldsymbol{p}_b(f(x; \widetilde{\boldsymbol{W}}^{(1:l)})) \\ \boldsymbol{p}_c(f(x; \boldsymbol{W}^{(1:l)})) - \boldsymbol{p}_c(f(x; \widetilde{\boldsymbol{W}}^{(1:l)})) \end{bmatrix}. \tag{3}$$

Given that various layers contribute to the model's performance in distinct ways Frankle et al. (2020), the impact of pruning layer weights would varies from layer to layer, particularly the varying information carried by active foreground points/voxels across layers. Assigning an appropriate layer-wise sparsity level for each layer could significantly impact performance. In this regard, our proposed pruning problem is formulated to obtain a layer-wise sparsity allocation that minimizes both bounding box localization distortion and confidence score distortion, constrained to a specified computation reduction target (FLOPs). Hence we formulate a pareto-optimization problem:

$$\min.\, E\left(\left\|\boldsymbol{\lambda}^\top (\boldsymbol{y} - \tilde{\boldsymbol{y}})\right\|^2\right) \quad s.t.\, \frac{\text{FLOPs}(f(\widetilde{\boldsymbol{W}}^{(1:l)}))}{\text{FLOPs}(f(\boldsymbol{W}^{(1:l)}))} \leq R, \tag{4}$$

which jointly minimizes the distortion caused by pruning under a certain FLOPs reduction target $R$. $\boldsymbol{\lambda} \in \mathbb{R}^2_+$ is the Lagrangian multiplier.

### 4.2 SECOND-ORDER APPROXIMATION OF DETECTION DISTORTION

In order to obtain a tractable solution, the above objective needs to be transformed into some closed-form functions of the optimization variable, which is the layer-wise pruning ratio. For each layer, given a parameter scoring method, the corresponding pruning error on weight $\Delta \boldsymbol{W}$ is also determined. First, we expand the distortion $\boldsymbol{y} - \tilde{\boldsymbol{y}}$ using the second-order Taylor expansion (We omit the superscript $(1:l)$ for visual clarity from now)

$$\boldsymbol{y} - \tilde{\boldsymbol{y}} = \sum_{i=1}^l \frac{1}{2} \Delta \boldsymbol{W}^{(i)\top} \boldsymbol{H}_i \Delta \boldsymbol{W}^{(i)}, \tag{5}$$

---

**Algorithm 1** Optimization via dynamic programming.

---

**Input:** $T$: The total number of weights to be pruned. $\delta_{i,k}$: Output distortion when pruning weights in layer $i$, for $1 \le i \le l$ and $1 \le k \le T$.
**Output:** The layerwise pruning ratios $\alpha_i^*$, for $1 \le i \le l$.
  **for** $i$ from 1 to $l$ **do**
    **for** $j$ from 0 to $T$ **do**
      If $i = 1$: $g_1^j \leftarrow \delta_{1,j}$, $s_1^j \leftarrow j$.
      Else: $g_i^j \leftarrow \min\{g_{i-1}^{j-k} + \delta_{i,k}\}$, $s_i^j \leftarrow \arg\min_k\{g_i^j\}$.
    **end for**
  **end for**
  **for** $i$ from $l$ to 1 **do**
    The number of weights pruned in layer $i$ is $s_i^T$.
    The pruning ratio of layer $i$ is $\alpha_i^* = \frac{s_i^T}{|\boldsymbol{W}^{(i)}|}$
    Update $T \leftarrow T - s_i^T$.
  **end for**

---

where $\boldsymbol{H}_i$ is the Hessian matrix of the $i$-th layer weight. We discard the first-order term since it has been shown neglegible on pre-trained models by prior studies.

Then consider the expectation of the squared L2 norm in the objective Eq. 4, which can be rewritten as the vector inner-product form:

$$E(\|\boldsymbol{\lambda}^\top(\boldsymbol{y} - \tilde{\boldsymbol{y}})\|^2) = \sum_{i,j=1}^{l} E\left[\left(\frac{1}{2}\Delta\boldsymbol{W}^{(i)\top}\boldsymbol{H}_i\Delta\boldsymbol{W}^{(i)}\right)^\top \boldsymbol{\lambda}\boldsymbol{\lambda}^\top\left(\frac{1}{2}\Delta\boldsymbol{W}^{(j)\top}\boldsymbol{H}_j\Delta\boldsymbol{W}^{(j)}\right)\right].$$

(6)

When we further expand the inner-product term, the cross-term for each layer pair $(i, j)$ ($1 \le i \ne j \le l$) is:

$$E\left[\frac{1}{4}\Delta\boldsymbol{W}^{(i)}\Delta\boldsymbol{W}^{(i)\top}\boldsymbol{H}_i^\top\boldsymbol{\lambda}\boldsymbol{\lambda}^\top\Delta\boldsymbol{W}^{(j)\top}\boldsymbol{H}_j\Delta\boldsymbol{W}^{(j)}\right].$$

(7)

When we discuss the influence of the random variable $\Delta\boldsymbol{W}$, we can treat the first-order and second-order derivatives $\nabla_{\boldsymbol{W}}\boldsymbol{y}$ and $\boldsymbol{H}$ as constants and thus move them out of expectation. Also vector transpose is agnostic inside expectation. Then Eq. 7 becomes

$$\frac{1}{4}\boldsymbol{H}_i^\top\boldsymbol{\lambda}\boldsymbol{\lambda}^\top\boldsymbol{H}_j E(\|\Delta\boldsymbol{W}^{(i)\top}\Delta\boldsymbol{W}^{(j)}\|^2).$$

(8)

Using Assumption 1, we can find that the above cross-term are also equal to zero, then we can derive the expectation of the distortion as follows.

$$E(\|\boldsymbol{\lambda}^\top(\boldsymbol{y} - \tilde{\boldsymbol{y}})\|^2) = \sum_{i=1}^{l} E\left(\left\|\frac{1}{2}\boldsymbol{\lambda}^\top\Delta\boldsymbol{W}^{(i)\top}\boldsymbol{H}_i\Delta\boldsymbol{W}^{(i)}\right\|^2\right).$$

(9)

After the above relaxation, we estimate the original objective as:

$$\text{min.} \sum_{i=1}^{l} E\left(\left\|\frac{1}{2}\boldsymbol{\lambda}^\top\Delta\boldsymbol{W}^{(i)\top}\boldsymbol{H}_i\Delta\boldsymbol{W}^{(i)}\right\|^2\right) \quad s.t. \frac{\text{FLOPs}(f(\widetilde{\boldsymbol{W}}^{(1:l)}))}{\text{FLOPs}(f(\boldsymbol{W}^{(1:l)}))} \le R.$$

(10)

### 4.3 Optimization Stratgy and Empirical Complexity Analysis

Let us denote $\alpha_{i,k}$ to represent the pruning ratio at layer $i$ by $k$ weights, where $0 \le \alpha_{i,k} \le 1$ for all $i$ and $k$. In addressing the pruning problem defined by Eq. 10, our approach involves selecting the optimal pruning ratios to minimize the distortion as expressed in Eq. 10. Denote the term inside L2 norm in Eq. 10 as $\delta$:

$$\delta = \frac{1}{2}\boldsymbol{\lambda}^\top\Delta\boldsymbol{W}^{(i)\top}\boldsymbol{H}_i\Delta\boldsymbol{W}^{(i)}.$$

(11)

Suppose we have obtained a set of $\delta_{i,k}$, where $\delta_{i,k}$ represents the distortion error when pruning $k$ weights at layer $i$. Specifically, define $g$ as the state function, in which $g_i^j$ means the minimal distortion caused when pruning $j$ weights at the first $i$ layers. The searching problem can be addressed by decomposing the original problem into sub-problems by the following state translation rule:

$$g_i^j = \min\{g_{i-1}^{j-k} + \delta_{i,k}\}, \ where \ 1 \leq k \leq j. \tag{12}$$

We achieve the optimal pruning solution by employing dynamic programming using the translation rule, as outlined in Algorithm 1, with linear time complexity relative to model parameter size.

### 4.4 Performance Optimization and Empirical Complexity

**Hessian approximation.** For empirical networks, we approximate the hessian matrix $\boldsymbol{H}_i$ using *empirical Fisher* $\hat{\boldsymbol{F}}$ Kurtic et al. (2022):

$$\boldsymbol{H}_i = \boldsymbol{H}_{\boldsymbol{y}}(\boldsymbol{W}^{(i)}) \approx \hat{\boldsymbol{F}}(\boldsymbol{W}^{(i)}) = \kappa \boldsymbol{I}_d + \frac{1}{N} \sum_{n=1}^{N} \nabla_{\boldsymbol{W}^{(i)}} \boldsymbol{y}_n \nabla_{\boldsymbol{W}^{(i)}}^{\top} \boldsymbol{y}_n, \tag{13}$$

where $\kappa \geq 0$ is a small dampening constant, $I_d$ is the identity matrix. A straightforward way to compute $\delta_{i,k}$ on a calibration set of size $N$ is to iterate through different pruning ratios $\alpha_{i,k}$ to determine the corresponding $\delta_{i,k}$ values. However, even with the use of the approximated Hessian, the process would still be computationally intensive at the complexity of $O(NKD_i^4)$, where $K$ is the number of possible pruning ratios and $D_i = |\boldsymbol{W}^{(i)}|$ is the number of neurons in $i$-th layer weight. This poses challenge in enhancing the efficiency of the proposed method efficient to fully leverage the advantages of a sparse network. We observe that the derivative $\nabla_{\boldsymbol{W}_i} \boldsymbol{y}$ remains constant with changes in the pruning ratio. This allows us to reuse the Hessian matrix for all pruning ratios, reducing the complexity to $O((N+K)D_i^2 + KD_i^4)$. However, the existence of the biquadratic complexity makes it still excessively costly. Additionally, we observe that with a slight increase in the pruning ratio, only a small subset of neurons is identified for further removal from the weight tensor that has already undergone pruning. Specifically, this means if we denote $\Delta \boldsymbol{W}_{i,k}^{(i)}$ as the difference on weight tensor of $i$-th layer caused by pruning with the pruning ratio of $\alpha_{i,k}$, the $\sigma_{i,k} = \Delta \boldsymbol{W}_{i,k}^{(i)} - \Delta \boldsymbol{W}_{i,k}$ will be very sparse each time when pruning ratio increases from $\alpha_{i,k-1}$ to $\alpha_{i,k}$. We select a subvector $\text{sp}(\sigma_{i,k})$ from $\sigma_{i,k}$ which has much less dimensions than $\sigma_{i,k}$, where $\text{sp}(\cdot)$ denotes the sparsification operation to extract non-zero values from vector. Therefore, we can leverage this observation to vastly reduce the calculation by only calculate the incremental value only related to the subvector $\text{sp}(\sigma_{i,k})$, if we have calculated the distortion value of the previous step. We can then select certain rows in $\nabla_{\boldsymbol{W}^{(i)}}$ in Eq. (13) that are visible to $\text{sp}(\sigma_{i,k})$ for the subsequent calculations in Eq. (11) to obtain a reduced hessian $\boldsymbol{H}_i'$ without affecting the calculations results. We derive the following update rule to efficiently obtain $\delta_{i,k}$ given the $\delta_{i,k-1}$:

$$\delta_{i,k} - \delta_{i,k-1} = \underbrace{\frac{1}{2} \text{sp}(\sigma_{i,k})^{\top} \boldsymbol{H}_i' \text{sp}(\sigma_{i,k})}_{①} + \underbrace{\text{sp}(\Delta \boldsymbol{W}_{i,k-1})^{\top} \boldsymbol{H}_i' \text{sp}(\sigma_{i,k})}_{②}. \tag{14}$$

Denote the dimension of the subvector $\sigma^{(i)}(\alpha_{i,k})$ as $d_{i,k} \ll D_i$ equals the number of neurons newly pruned within $\Delta \boldsymbol{W}^{(i)}(\alpha_{i,k-1})$ compared to $\Delta \boldsymbol{W}^{(i)}(\alpha_{i,k})$ as pruning ratio $\alpha_{i,k}$ increases from $\alpha_{i,k-1}$, the multiplication calculation in Eq. 14 can be operated at lower dimensions, where $\nabla_{\boldsymbol{W}^{(i)}}^{\top\prime} \boldsymbol{y} \in \mathbb{R}^{d_{i,k}}, \boldsymbol{H}_i' \in \mathbb{R}^{D_i \times d_{i,k}}$ are subvector and submatrix indexed from the original ones. To further elinimate any potential confusion, we illustrate the above update rule Eq. 14 in Fig. 3 in the Appendix. Given that $\alpha_{i,0} = 0$, indicating no pruning at all, this ensures that $\delta_{i,0} = 0$. Therefore, the complexity becomes the summation of $K-1$ times of updating $O(\sum_{k=2}^{K} d_{i,k} \sum_{k'=1}^{k-1} d_{i,k'})$. Since $\alpha_{i,k}$ increases linearly, the $d_{i,k} \approx \frac{D_i}{K}$, therefore, the complexity is around $O(\frac{N}{2} D_i^2)$. Hence the total computation complexity for calculating the distortion $\delta_{i,k}$ across all $l$ layers is around $O(\frac{1}{2} \sum_{i=1}^{l} D_i^2)$, significantly lower than the original complexity.

By far, we established a weight pruning framework that is formulated to minimize distortion in 3D object detection. Algorithm 2 describes the holistic pruning procedure of the DM3D method.

Table 1: Performance comparison of DM3D on ONCE *val* set. Gray background indicates dense model results. For baseline sparse detection results, we list the performance drop with their corresponding dense ones reported in their original papers.

| Method | FLOPs (%) | mAP (drop) | Vehicle (IoU=0.7) | | | Pedestrian (IoU=0.5) | | | Cyclist (IoU=0.5) | | |
|---|---|---|---|---|---|---|---|---|---|---|---|
| | | | 0-30 | 30-50 | 50-Inf | 0-30 | 30-50 | 50-Inf | 0-30 | 30-50 | 50-Inf |
| PointRCNN Shi et al. (2019) | / | 28.74 | 52.09 | - | - | 4.28 | - | - | 29.84 | - | - |
| PointPillar Lang et al. (2019) | / | 44.34 | 68.57 | - | - | 17.63 | - | - | 46.81 | - | - |
| SECOND Yan et al. (2018) | / | 51.89 | 71.16 | - | - | 26.44 | - | - | 58.04 | - | - |
| PVRCNN Shi et al. (2020) | / | 52.44 | 87.54 | 72.29 | 57.22 | 21.91 | 20.89 | 18.18 | 69.8 | 54.16 | 36.8 |
| MultiLi et al. (2023) | 60.61 | - | −2.85 | −4.42 | −2.81 | −5.89 | −6.76 | −0.58 | −10.73 | −8.12 | −4.77 |
| DM3D (Ours) | 60.61 | +2 | +0.1 | **−2.07** | **−2.55** | **+11.61** | **+5.41** | **+0.41** | **+0.31** | **−0.25** | **−0.91** |
| SECOND Yan et al. (2018) | / | 51.43 | 83.28 | 67.13 | 49.82 | 26.65 | 22.88 | 15.58 | 68.69 | 52.06 | 33.3 |
| Multi Li et al. (2023) | 52.54 | - | −1.49 | −5.35 | −4.03 | −7.84 | −6.24 | −2.98 | −13.32 | −8.83 | −4.69 |
| DM3D (Ours) | 52.54 | −1.6 | **0.45** | **−1.75** | **0.0** | **−2.44** | **−3.91** | **−1.54** | **−1.53** | **−2.55** | **−0.18** |
| CenterPoint Yin et al. (2021) | - | 64.01 | 76.09 | - | - | 49.37 | - | - | 66.58 | - | - |
| Ada3D Zhao et al. (2023) | 26.82 | −1.31 | −2.26 | - | - | −0.71 | - | - | −0.95 | - | - |
| DM3D (Ours) | 26.82 | **−0.7** | **−0.71** | - | - | **−0.48** | - | - | **−0.94** | - | - |

Table 2: Performance comparison of DM3D on nuScenes *val* set.

| Method | FLOPs (%) | mAP (drop) | NDS (drop) |
|---|---|---|---|
| PointPillar Lang et al. (2019) | / | 44.63 | 58.23 |
| SECOND Yan et al. (2018) | / | 50.59 | 62.29 |
| CenterPoint-Pillar Yin et al. (2021) | / | 50.03 | 60.70 |
| CenterPoint (*voxel=0.1*)Yin et al. (2021) | / | 55.43 | 64.63 |
| Ada3DZhao et al. (2023) (*voxel=0.1*) | 33.24 | 54.8 (−0.63) | 63.53 (−1.1) |
| DM3D (Ours) (*voxel=0.1*) | 33.24 | **55.32 (−0.11)** | **64.36 (−0.27)** |
| VoxelNeXT Chen et al. (2023b) | / | 60.5 | 66.6 |
| Ada3D Zhao et al. (2023) | 85.12 | 59.75 (−0.75) | 65.84 (−0.76) |
| DM3D (Ours) | 85.12 | **60.91 (+0.41)** | **66.91 (+0.31)** |

Table 3: Performance comparison of DM3D on KITTI *val* set for Car class.

| Method | FLOPs (%) | Easy (drop) | Mod. (drop) | Hard (drop) | FLOPs (%) | Easy (drop) | Mod. (drop) | Hard (drop) |
|---|---|---|---|---|---|---|---|---|
| | Voxel R-CNN Deng et al. (2021) | | | | SECOND Yan et al. (2018) | | | |
| Dense | / | 89.44 | 79.2 | 78.43 | / | 88.08 | 77.77 | 75.89 |
| SPSS-Conv Liu et al. (2022a) | 73.0 | **+0.28** | +0.05 | −0.04 | 88.31 | +0.21 | −0.11 | −0.15 |
| DM3D (Ours) | 74.36 | +0.04 | **+0.06** | **+0.11** | 78.38 | **+0.10** | **+0.11** | **−0.03** |

## 5 EXPERIMENTS

### 5.1 EXPERIMENTAL SETTINGS

**Datasets and Models.** We evaluate three 3D object detection tasks to prove the effectiveness of our proposed method, namely KITTI, nuScenes and ONCE. KITTI dataset includes 3712 training examples, 3769 validation examples and 7518 test examples. Detection targets are categorized into three classes: Car, Pedestrian, and Cyclist, with ground truth bounding boxes divided into "Easy", "Moderate", and "Hard" difficulty levels. Evaluation of detection performance employs average precision (AP) for each category, using an IoU threshold of 0.7 for cars and 0.5 for pedestrians and cyclists. The nuScenes dataset is a comprehensive autonomous driving dataset containing 1,000 driving sequences with different modalities including LIDAR and cameras. We follow the default split, where *train* split includes 700 training scenes and *val* set has 150 scenes. The ONCE dataset is a large-scale LiDAR-based collection for autonomous driving, featuring 1 million scenes with 16k fully annotated for 3D object detection, utilizing mAP for performance evaluation. We perform post-train pruning followed by one round of finetuning to fully recover the performance. We provide implementation details in the Appendix A.2.1.

**FLOPs Calculation.** Since we prune both 3D and 2D backbones from the detection model, we accordingly report the FLOPs of sparsified layers in both 3D and 2D backbones. For baselines that only perform voxel sparsification in 3D backbones, we recalculate the FLOPs reduction w.r.t. 3D and 2D backbones for fair comparison. Except for results in Tab. 4, when analyzing the effect of pruning the detection head, we report the FLOPs w.r.t. the whole network for all three cases.

## 5.2 MAIN RESULTS

We conduct extensive evaluations of our pruning method on various 3D object detection datasets including Nuscene, ONCE and KITTI. As we presented in Tab. 1, on ONCE validation dataset, DM3D achieves **higher** detection precision for **all** three tested detectors, PVRCNN Shi et al. (2020), SECOND Yan et al. (2018) and CenterPoint Yin et al. (2021) compared to the sparse baseline methods under the same FLOPs reduction level. On PVRCNN Shi et al. (2020) and SECOND Yan et al. (2018) model, we outperform the voxel pruning scheme Li et al. (2023) on all precision metrics of class Car Pedestrian and Cyclist. We also observe a **huge performance boost** from the baseline dense model on PVRCNN on the mAP score by $2\%$. On CenterPoint Yin et al. (2021), we also outperform the current SOTA Zhao et al. (2023). In Tab. 2, we present the results on nuScenes dataset. Again, we witness less performance drop at the same level of FLOPs reduction with the baseline w.r.t. to both mAP and NDS metrics. On the recent VoxelNeXT network, compared to the Ada3D Zhao et al. (2023) with a $0.75$ mAP drop, we boost the mAP on the pruned model by $0.41$. Comparisons on KITTI datasets are shown in Tab. 3. Our method continues to achieve on-par performances with baselines on two prevailing models especially for Car AP Mod. score. On Voxel R-CNN, SPSS-Conv Liu et al. (2022a) performs extremely well on Car AP Easy with a $0.28$ gain from dense Voxel R-CNN, whereas our DM3D method brings slightly less but still a positive performance gain.

We demonstrate our overall detection performance on one of the datasets in Fig. 1. We compare ourselves with two baselines Multi Li et al. (2023) and Ada3D Zhao et al. (2023) on three detection networks with different FLOPs reduction levels. On CenterPoint, we compared the pareto-frontier of our pruning scheme to Ada3D, the only literature reporting the mAP performance. Using our DM3D scheme, we achieve a $3.89\times$ FLOPs reduction from original CenterPoint model while outperforming Ada3D. On SECOND and PVRCNN, we perform consistently better than baselines on different detection categories while gradually decreasing the FLOPs and achieves around $2\times$ speedup.

## 5.3 ABLATION STUDY AND DISCUSSIONS

**Controlling pruning rates of different parts in backbone.** As shown in Tab. 4, we explore the possibility of pruning more and more redundant weights from different modules of the detection models. Start from pruning 3D backbone only, we prune out $47.62\%$ FLOPs from 3D backbone of SECOND model, resulting in a total FLOPs reduction of $93.2\%$ w.r.t. the whole network. Then we prune the 3D and 2D backbone together, keeping the detection head untouched. Finally, we also include head layers into the pruning, resulting in a total FLOPs reduction of $64.9\%$. Intuitively, we notice that the AP scores gradually decrease as the total FLOPs decreases, but by very marginal amounts, showing that there is still a **large degree of weight redundancy** in 3D detection model.

Table 4: Ablation study when pruning only certain parts of model of SECOND on KITTI *val* dataset.

| 3D (%) | 2D (%) | Head (%) | FLOPs (%) | Car AP | | | Ped. AP | | | Cyc. AP | | |
|---|---|---|---|---|---|---|---|---|---|---|---|---|
| | | | | Easy | Mod. | High | Easy | Mod. | High | Easy | Mod. | High |
| / | / | / | / | 88.09 | 77.77 | 75.91 | 53.43 | 48.63 | 44.2 | 81.8 | 66.04 | 62.47 |
| 47.62 | 100 | 100 | 93.2 | +0.14 | +0.17 | +0.06 | −0.42 | −0.77 | −0.39 | −0.2 | −0.33 | −0.03 |
| 47.62 | 67.57 | 100 | 79.22 | −0.41 | −0.33 | −0.52 | +0.5 | −0.22 | −0.12 | −1.65 | −1.61 | −1.22 |
| 47.62 | 67.57 | 67.37 | 64.9 | −0.43 | −0.1 | −0.37 | +0.15 | +0.16 | −0.62 | −1.78 | −1.45 | −1.22 |

**Second-order Distortion *v.s.* True Distortion.** To evaluate whether the proposed hessian-based distortion approximation scheme is faithful to the actual detection distortion from network output, we conducted an experiment on KITTI *val* dataset using the distortion data $\delta_{i,k}$ from real network prediction $y$ on calibration set as originally described in Eq. 4. Tab. 5 shows no significant difference between the two approaches, verifying the effectiveness of the proposed Hessian approximation.

Table 5: Comparison of the proposed hessian-based pruning scheme with pruning using distortion $\delta_{i,k}$ from actual network output.

| Method | FLOPs (%) | Car AP drop | | | Ped. AP drop | | | Cyc. AP drop | | |
|---|---|---|---|---|---|---|---|---|---|---|
| | | Easy | Mod. | High | Easy | Mod. | High | Easy | Mod. | High |
| SECOND Yan et al. (2018) | / | 88.09 | 77.77 | 75.91 | 53.43 | 48.63 | 44.2 | 81.8 | 66.04 | 62.47 |
| Hessian (DM3D) | 87.85 | +**0.14** | +**0.17** | +**0.06** | −0.42 | −**0.77** | −0.39 | −**0.2** | −0.33 | −**0.03** |
| Actual Dist. | 87.85 | −0.26 | +0.05 | +**0.1** | +**0.51** | −0.9 | −**0.12** | −0.78 | +**0.48** | −0.35 |

**Ablation study of different $\lambda$.** We explored the impact of different $\lambda$ in Eq. (10) in Tab. 6. For $\lambda = [\lambda_1, \lambda_2]^\top$, we keep $\lambda_1 = 1$ and range $\lambda_2$ from 4 to 0.5. We notice that different choices of the $\lambda$ do marginal impact on the overall detection performances, meanwhile some object classes favors certain choices of the $\lambda$ scattering around 2 to 0.5 to limited extend. This means our method are mostly robust to the choice of $\lambda$ and we simply fix $\lambda$ as $[1, 1]^\top$ in all the experiments for the comprehensive performance of the pruned models.

Table 6: Ablation studies of different head box and classication ratio $\lambda$ on Kitti *val* set. The best results for each metrics are marked in bold and the second bests are marked with underline.

| Method | FLOPs (%) | $\lambda_2/\lambda_1$ | Car (IoU=0.7) easy | moderate | hard | Pedestrian (IoU=0.5) easy | moderate | hard | Cyclist (IoU=0.5) easy | moderate | hard |
|---|---|---|---|---|---|---|---|---|---|---|---|
| SECOND | / | / | 88.09 | 77.77 | 75.91 | 53.43 | 48.63 | 44.20 | 81.80 | 66.04 | 62.46 |
| DM3D(Ours) | 87.85 | 4 | −0.70 | −0.34 | −0.41 | **−0.07** | −0.35 | −0.31 | −1.2 | −0.95 | −0.43 |
| DM3D(Ours) | 87.85 | 2 | −0.19 | −0.13 | −0.24 | −0.36 | **−0.08** | **−0.07** | **−0.32** | **−0.25** | **+0.45** |
| DM3D(Ours) | 87.85 | 1 | −0.20 | −0.14 | −0.09 | −0.23 | −1.26 | −0.71 | −0.54 | −0.32 | −0.02 |
| DM3D(Ours) | 87.85 | 0.5 | **+0.01** | **−0.09** | **+0.03** | −0.40 | −0.44 | −0.65 | −0.60 | −0.53 | +0.10 |

**Ablation study of different $K$.** Fig. 2 shows the effects of different $K$ (granularity of pruning ratio sampling) from the perspective of both performance and pruning overhead. It shows that as $K$ increases, while the detection scores drop slightly from dense models, pruning overhead rapidly increases. In practice, we use $K = 1000$ for optimal cost-effective pruning performance.

We include more discussions in the Appendix A.2, including qualitative results and discussions on comparisons against layer-wise pruning counterparts.

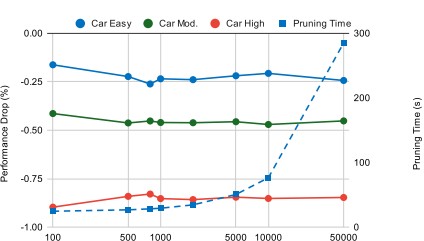

Figure 2: The effect of different K values on detection performance and pruning overhead. X-axis is in log-scale.

### 5.4 COMBINATION WITH SPATIAL PRUNING

Our proposed weight pruning scheme works orthogonally with the prevailing spatial pruning approaches. To assess the possibility of uniting these approaches to reduce computations, we employ the DM3D weight pruning on top of spatial pruning method SPSS-Conv Liu et al. (2022a) that prunes out $50\%$ of redundant voxels, and further prunes out parameters from 3D and 2D Backbones with a FLOPs reduction of $50.66\%$ w.r.t. the whole network. As shown in Tab. 7, we observe that pruning weights from spatially sparse network brings negligible performance drop using DM3D, while the speedup is boosted from $1.36\times$ to $1.97\times$, showing the generalizability of DM3D with standard spatial pruning methods.

Table 7: Combining spatial and weight sparsity scheme on KITTI *val* dataset.

| Method | Sparsity (%) Voxel | Weight | Total FLOPs (%) | Easy (drop) | Mod. (drop) | Hard (drop) |
|---|---|---|---|---|---|---|
| Voxel R-CNN Deng et al. (2021) | 100 | 100 | 100 | 89.44 | 79.2 | 78.43 |
| SPSS-Conv Liu et al. (2022a) | 50 | 100 | 73 | +0.28 | +0.05 | −0.04 |
| SPSS-Conv Liu et al. (2022a) + DM3D | **50** | **50.8** | **50.66** | **−0.04** | **+0.08** | **+0.02** |

## 6 CONCLUSIONS

We have presented a weight pruning scheme for voxel-based 3D object detection models orthogonal to prevailing spatial redundancy-based approaches. The pruning scheme is based on second-order Taylor approximation on the detection distortion, which is able to minimize detection locality and confidence degradation on pruned model. The scheme is extremely lightweight, with polynomial complexity for Hessian information acquisition and linear complexity for layerwise sparsity search. We show the superiority of the novel scheme on various 3D detection benchmarks over state-of-the-art approaches that exploits only spatial redundancy. In the future, we aim to develop a unified pruning scheme that leverages both weight and spatial redundancies for optimal results.

## 7 REPRODUCIBILITY STATEMENT

We pay attention to the reproducibility of this work. We provided details of the implementation details in the appendix (see Appendix A.2.1).

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

## A  APPENDIX

### A.1  DETAILS OF ALGORITHMS

#### A.1.1  HESSIAN REDUCED CALCULATION SCHEME.

We illustrate the details of the abovementioned vector and matrix notations as well as the steps of calculation needed to obtain second-order distortion for $\alpha_k$ given the distortion of $\alpha_{k-1}$ as in Fig. 3.

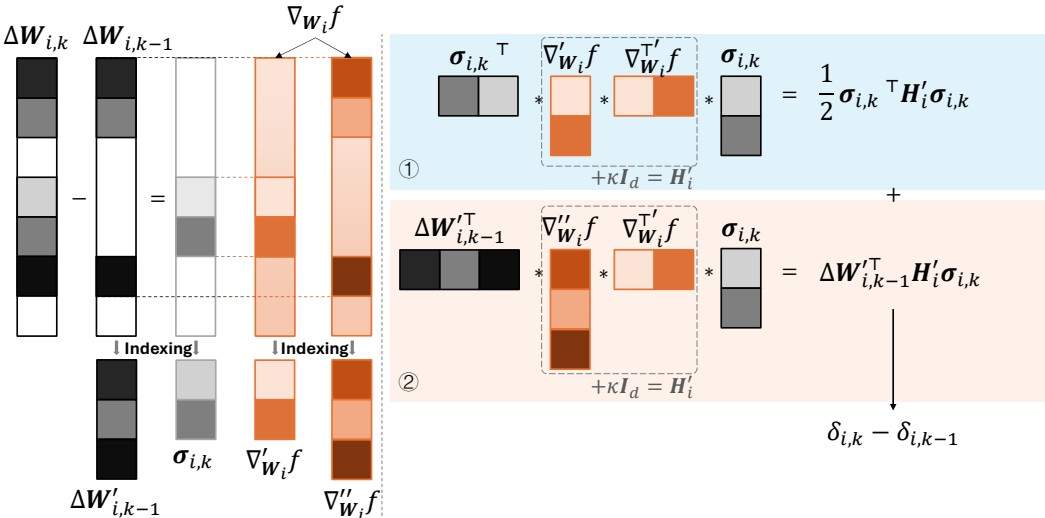

Figure 3: Detailed reduced calculation of $\delta_{i,k}$ described in Eq. 14.

#### A.1.2  END-TO-END METHOD.

We summarize the holistic algorithm for DM3D in the pseudocode in Algorithm 2.

### A.2  MORE EXPERIMENTAL DISCUSIONS

#### A.2.1  IMPLEMENTATION DETAILS.

We use OpenPCDet Team (2020) framework to perform fine-tuning. We use pre-trained weights from official sites as possible for post-train pruning. For others, we use our retained model in KITTI and ONCE which train 80 epochs with all default setting. We set the fine-tuning batch size as 64 on 8 A100 40GB GPUs with Adam optimizer, weight decay of 0.01 for all experiments. We generally set the fine-tune learning rates $10\times$ lower than the dense training learning rates. We set 30 epoch for KITTI and nuScenes, 60 epoch for ONCE. We search all the fine-tune models and find the best performance based on the metric(describe above) of each dataset.

#### A.2.2  COMPARISONS WITH LAYER-WISE WEIGHT PRUNING METHODS

To see how competitive our layer-wise weight pruning method is compared to other counterparts, we conduct extensive experiments on the dataset KITTI, as shown in Tab. 8. We can see that our scheme yields the best performance on most of the metrics, expect for one metric where Uniform+ Gale et al. (2019) is slightly better.

#### A.2.3  QUALITATIVE RESULTS.

In Fig. 4, we visualize the qualitative performance of our pruning scheme applied to VoxelNeXT on nuScenes dataset, which generates 60.91 mAP. One can observe that 3D detector pruned by our DM3D approach generates high quality bounding boxes for essential visual categories in LiDAR data, further verifying the effectiveness of the DM3D.

---

**Algorithm 2** Distortion-Minimized Pruning of 3D Object Detection Model.

---

**Input:** Training dataset $\mathcal{D}_t$, Calibration dataset $\mathcal{D}_c$, 3D detection model $\mathcal{F}$ with $l$ layers, Number of possible pruning ratios for each layer $K$, Fine-tuning epochs $E$.

**Output:** The pruned 3D detection model $\tilde{\mathcal{F}}$.

    Inference $\mathcal{F}$ on $\mathcal{D}_c$ to get output detections: $\mathbb{Y} \leftarrow \{\boldsymbol{y}(X) \mid \forall X \in \mathcal{D}_c\}$.

    Perform back-propagation on $\mathcal{F}$.

    Get a list of averaged gradients of each layers: $\mathbb{G} = \{\nabla_{\boldsymbol{W}^{(i)}} \boldsymbol{y} \mid 1 \le i \le l\}$.

    **for** $i$ from 1 to $l$ **do**

        $\alpha_{i,0} \leftarrow 0, \delta_{i,0} \leftarrow 0$.

        $\Delta \boldsymbol{W}^{(i)}(\alpha_{i,0}) \leftarrow \boldsymbol{0}$.

        **for** $k$ from 1 to $K$ **do**

            $\alpha_{i,k} \leftarrow \frac{k+1}{K}$.

            Prune $\boldsymbol{W}^{(i)}$ to get $\widetilde{\boldsymbol{W}}^{(i)}$ given $\alpha_{i,k}$: $\widetilde{\boldsymbol{W}}^{(i)} \leftarrow \boldsymbol{W}^{(i)} \odot \boldsymbol{M}_{\alpha_{i,k}}(\boldsymbol{S})$.

            Calculate pruning error matrix: $\Delta \boldsymbol{W}^{(i)}(\alpha_{i,k}) \leftarrow \boldsymbol{W}^{(i)} - \widetilde{\boldsymbol{W}}^{(i)}$.

            Calculate $\delta_{i,k}$ following Eq. 14.

        **end for**

    **end for**

    Obtain layerwise pruning ratios $\alpha_i^*$ using $\delta_{i,k}$ from Algorithm 1.

    **for** $i$ from 1 to $l$ **do**

        Prune $\boldsymbol{W}^{(i)}$ given $\alpha_i^*$: $\boldsymbol{W}^{(i)} \leftarrow \boldsymbol{W}^{(i)} \odot \boldsymbol{M}_{\alpha_i^*}(\boldsymbol{S})$.

    **end for**

    **for** $e$ from 1 to $E$ **do**

        Finetune $\tilde{\mathcal{F}}$ on $\mathcal{D}_t$.

    **end for**

---

Table 8: Performance comparison of DM3D with layer-wise CNNs weight pruning methods on KITTI *val* set for Car class.

| Method | *FLOPs* (%) | Easy (drop) | Mod. (drop) | Hard (drop) | *FLOPs* (%) | Easy (drop) | Mod. (drop) | Hard (drop) |
|---|---|---|---|---|---|---|---|---|
| | Voxel R-CNN Deng et al. (2021) | | | | SECOND Yan et al. (2018) | | | |
| Dense | / | 89.44 | 79.2 | 78.43 | / | 88.08 | 77.77 | 75.89 |
| LAMP Lee et al. (2020) | 74.0 | +0.46 | +0.05 | +0.99 | 78.0 | −0.58 | −0.77 | −1.16 |
| Global Morcos et al. (2019) | 74.0 | +0.36 | +0.13 | +0.93 | 78.0 | −0.52 | −0.34 | −0.50 |
| Uniform Zhu & Gupta (2018) | 74.0 | +0.47 | +0.26 | +1.00 | 78.0 | −0.48 | −0.37 | −0.61 |
| Uniform+ Gale et al. (2019) | 74.0 | **+0.69** | +0.32 | +0.99 | 78.0 | −0.74 | −0.47 | −0.83 |
| ERK Evci et al. (2020) | 74.0 | +0.48 | +0.37 | +1.11 | 78.0 | −0.38 | −0.61 | −0.72 |
| DM3D (Ours) | 74.95 | +0.59 | **+0.32** | +1.01 | 78.38 | **−0.47** | **−0.75** | **−1.34** |

### A.2.4 LAYER-WISE SPARSITY ALLOCATION RESULTS.

Fig. 5 shows the detailed allocation results of the proposed DM3D. In Fig. 5, we display the layer-wise sparsity levels optimized by DM3D under different FLOPs constraint levels. We analyze the behavior on three different networks. Since our method leverages weight redundancies, we are able to optimize the layer-wise sparsity of 3D and 2D backbones together and automatically decide the sparsity allocations. We observe that on PVRCNN, our scheme results in more sparsity in layers in 2D backbone than in 3D part, while on SECOND, it is the opposite case where more weights in 2D backbone is preserved. This implies that the expressiveness of PVRCNN is mostly coming from 3D feature extraction than SECOND. Earlier layers in 2D backbone survive from most pruning cases regardless of the networks, probably because these layers are essential to smoothly transfer information from 3D domain to 2D. As FLOPs target decreases, the sparsity distribution remains roughly the same, where most sensitive layers regarding to the detection distortion remain in less sparsity rates.

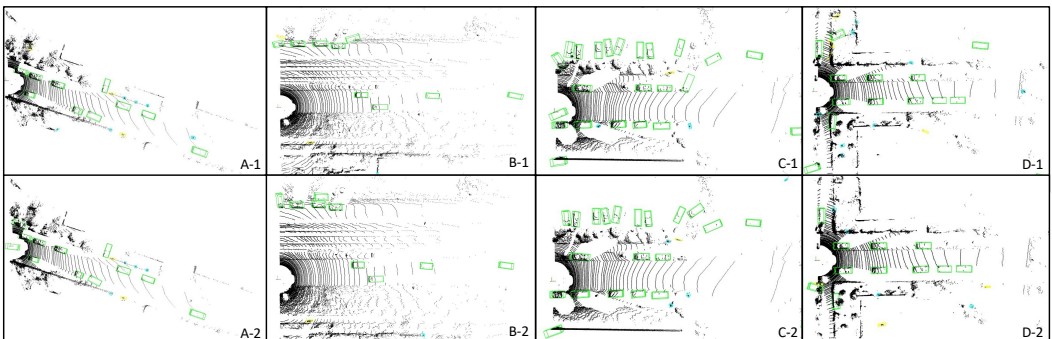

Figure 4: Qualitative performance of the pruned detection model on LiDAR data. "A", "B", "C", "D" refer to 4 scenes from KITTI dataset, of which "A" and "B" scenes are tested under the PVR-CNN model while "C" and "D" scenes are tested under the second model. "1" denotes the performance of model pruned by our method and "2" denotes the performance of the pre-trained model. For example, "A-1" denotes the performance of the PVRCNN-DM3D model, "A-2" denotes the performance of the Dense PVRCNN. This figure is best viewed by zoom-in.

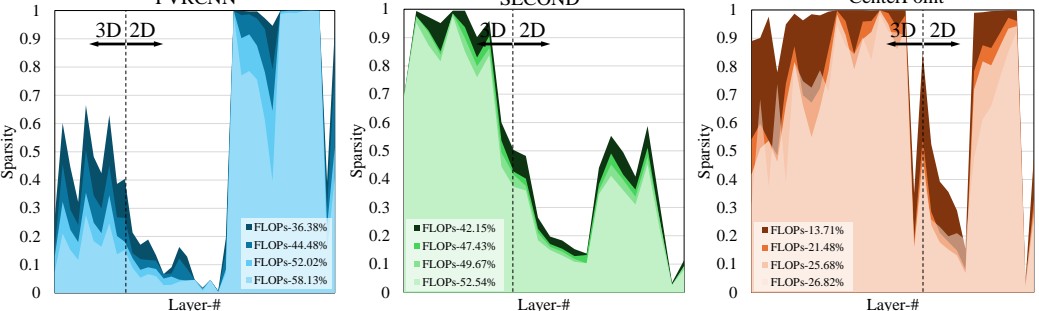

Figure 5: Layer-wise sparsity allocation on three detection networks on ONCE dataset. Sparsity level close to one means most weights getting pruned out.

