# OpenReview forum: "DM3D: Parameter-Efficient and Lossless 3D Object Detection via Distortion Minimization"
_ICLR.cc/2025/Conference — ICLR 2025 Conference Withdrawn Submission_

### Official Review · Reviewer_Wkq5 · 2024-10-15

**Soundness:** 3
**Presentation:** 2
**Contribution:** 3
**Rating:** 3
**Confidence:** 4

**Summary:**

This work introduces a novel post-training weight pruning technique tailored for 3D object detection. It distinguishes itself in two key aspects:

1. It operates independently of existing point cloud sparsification methods, focusing on redundant parameters in pre-trained models that have minimal impact on spatial precision and detection confidence (collectively referred to as “detection distortion”).

2. The method provides a flexible plug-and-play framework compatible with various sparsity strategies (including spatial sparsity) and any 3D detection model.

The DM3D leverages second-order Taylor approximation to identify layer-wise sparsity, minimizing detection distortion while significantly reducing model complexity without sacrificing accuracy. To efficiently manage the necessary second-order information, a lightweight algorithm is developed to collect Hessian data and use dynamic programming to optimize the layer-wise sparsity allocation.

**Strengths:**

1. Unlike common work that aims to improve detection performance, this paper focuses on how to lighten 3D detection, reduce memory requirements and speed up computation.

2. Overall, this work favours theory, and the experimental results are mainly aimed at improving previous methods. I think this paper is better aligned with the ICLR theme as work in the area of visual deep learning.

3. The work in this paper is more than adequate, with detailed content and reliable conclusions. It is recommended that parts of the supplementary material be added to the main text, which is limited to 10 pages by ICLR 2025.

**Weaknesses:**

1. The motivation of the introduction section is not clear enough. By illustrating the flaws of a large body of related work only shows that the research is worth following, but not why you designed the methods in the paper. It is recommended to illustrate the feasibility of rate distortion theory and weight pruning through theory or sample examples.

2. The methods section is almost entirely theoretical analysis or formula derivation, and some of the formulas are too repetitive. Examples include formulas (2-5) and related in-line formulas and text descriptions. This makes it difficult for the reader to associate the input point cloud $P$ and the 3D detector $D$ in this.

3. The experimental part is mainly based on classical detectors, however, they are all close to 5 years old work, e.g. PointRCNN, SECOND, CenterPoint. the latest about 3D detectors have been playing well in the multimodal domain, and their memory requirements will be much higher, it is recommended that the authors validate it on a newer 3D detector, which will be more convincing. Examples include PillarNeXt: Rethinking network designs for 3D object detection in LiDAR point clouds (CVPR 2023), Leveraging vision-centric multi-modal expertise for 3d object detection (NeurIPS 2024).

**Questions:**

How does the theoretical explanation of the method work on the 3D point cloud or 3D detector? It is recommended that the author give a clear explanation instead of just writing the theoretical derivation.

---

### Official Review · Reviewer_u4X9 · 2024-11-01

**Soundness:** 2
**Presentation:** 3
**Contribution:** 2
**Rating:** 3
**Confidence:** 4

**Summary:**

This work presents a post-training weight pruning technique specifically designed for voxel-based 3D object detection. This method is based on a second-order Taylor approximation of detection distortion to minimize the loss of detection locality and confidence in the pruned model. The use of a second-order Taylor approximation to identify layer-wise sparsity, allowing reduce model complexity. The results on different dataset and voxel-based 3D detection models demonstrate their effectiveness for weight pruning.

**Strengths:**

1. The weight pruning method is based on a second-order Taylor approximation, with detailed theoretical derivations and proofs.

2. The acquisition of Hessian information and layer-wise sparsity search have relatively low computational complexity.

3. This weight pruning method is orthogonal to spatial pruning methods.

**Weaknesses:**

1. The introduction appears to be unclear regarding the motivation; while I understand that pruning can reduce computational load and memory size, the authors seem to lack empirical observations or specific phenomena related to weight pruning in voxel-based models that could better support the motivation and inspiration for the design of this work. Maybe the author could provide specific examples or previous studies to demonstate it.

2. This pruning method appears to be applicable only to voxel-based point cloud detection methods; however, voxel-based approaches, through the implementation of sparse convolutions, can significantly reduce computational load and redundant calculations. Therefore, the applicability of this method seems somewhat limited. Ther author could discuss more potential possibility for non-voxel-based 3D objecte detection methods.

3. Using second-order Taylor  approximation for pruning/quantization error is already a common method for error approximation in model compression; thus, the methods described in section 4.2 are not particularly novel, and the overall innovation of the approach seems insufficient.

4. The authors should discuss the limitations of this method, such as computational overhead of the pruning process.

5.Lack some related works about quantization compression on 3D Point Cloud or Object Detection.
  (1.) LiDAR-PTQ: Post-Training Quantization for Point Cloud 3D Object Detection, In ICLR 2024.
  (2.) Bipointnet: Binary neural network for point clouds. In ICLR, 2021.

**Questions:**

1. FLOPs is merely a proxy for inference speed; a reduction in FLOPs does not equate to an improvement in model inference speed. The authors should report the actual inference latency of the pruned model to demonstrate the acceleration effect, rather than relying solely on FLOPs.

2. The authors should report the actual deployment time of this pruning method across different models, along with the corresponding inference acceleration effects of the pruned models.

3. Is the search time different for different layers?  Do the earlier or later layers generally have shorter search times?

4. In Table 1, the PVRCNN model shows a significant performance improvement for the pedestrian category after pruning; what is the reason for this?

5. How is the calibration set D_c selected for different datasets and models?

---

### Official Review · Reviewer_3pAF · 2024-11-02

**Soundness:** 2
**Presentation:** 2
**Contribution:** 2
**Rating:** 5
**Confidence:** 3

**Summary:**

This paper proposes a parameter-efficient method for 3D object detection. DM3D minimizes detection distortion using a second-order Taylor approximation and achieves significant FLOPs reductions with minimal performance impact, as demonstrated on KITTI, nuScenes, and ONCE datasets.

**Strengths:**

This paper introduces an innovative post-training weight pruning technique specifically designed for 3D object detection. Extensive experiments across multiple datasets demonstrate that the proposed method effectively reduces computational overhead with minimal impact on detection accuracy.

**Weaknesses:**

(1)This paper performs pruning operations on the network and compares FLOPS. From the perspective of parameter quantity and FLOPS, it does compress a lot. We are curious whether this method can compress the size of the network weight file significantly? With the reduction in FLOPS, how much will the inference speed of the network ultimately improve?

(2)In terms of spatial pruning, only a combination with SPSS Conv is evaluated. Exploring the effectiveness of this method in combination with Ada3D or other spatial pruning approaches would help illustrate its potential to complement existing techniques.

(3)Some issues in writing detract from the overall rigor. For instance, “Hard” is labeled as “high” in Tables 4, 5, and Figure 2, and “easy,” “mod,” and “hard” are presented in lowercase in Table 6, inconsistent with their notation in other tables.

**Questions:**

Why are there no specific accuracy results for each distance in the last three rows of Table 1? Additionally, the reported Mod. AP for Voxel-RCNN in Table 3 appears inconsistent with results from the original Voxel-RCNN paper and SPSS paper, which may require clarification.

---

### Official Review · Reviewer_PUbk · 2024-11-03

**Soundness:** 3
**Presentation:** 3
**Contribution:** 3
**Rating:** 5
**Confidence:** 5

**Summary:**

This paper presents a novel achitecture named DM3D, aiming for efficient 3d object detecion task. The authors presents a novel post-training weight pruning technique tailored specifically for 3D object detection. The proposed method stand out in two key ways. 1. It operates independently from existing point cloud sparsi- fication methods, targeting redundant parameters in pre-trained models that mini- mally affect both spatial accuracy and detection confidence. 2. it provides a flexible, plug-and-play framework compatible with other sparsity schemes including spatial sparsity and with any 3D detection model.

**Strengths:**

1. DM3D propose a generic weight pruning frame work for 3D object detection models capable of reducing computation complexity (FLOPs) via sparsifying weights.
2. The proposed method adopts a hessian-based layer-wise pruning scheme. Through performing dynamic programming and fine-grained optimization, the authors derive an extremely efficient al- gorithm with polynomial time complexity to find the global optimal solution.
3. The proposed method can be easily adapt to any Conv-based 3D object detector.
4. The paper is well organized which is easy to follow.

**Weaknesses:**

1. In the results of Table 1, many metrics for the baseline method are missing. I understand that the original authors may not have provided these results in the paper, but for a fair comparison, you are obligated to reproduce these results.
2. In Table 1, why does your method show a significant increase in the Prdestrain metric compared to PVCNN? On the other hand, Second and CenterPoint, which use the same 3D backbone, do not exhibit the same phenomenon. I am curious about the source of this increase, such as a different 2D head architecture.
3. The experiments are not comprehensive enough. Both SPS-Conv and Ada3d have results on nuScenes and KITTI datasets, but you only compared against Ada3d on nuScenes and only compared against SPS-Conv on the KITTI dataset.
4. I am very interested in the performance of your method on larger datasets, such as Waymo and Argoverse. These two datasets are important benchmarks for assessing the effectiveness of methods.
5. GFLOPS does not represent the actual model runtime speed. I am more interested in how your method compares in terms of runtime speed with other methods, as this will impact the final score of this article.

**Questions:**

Please refer to the weakness.

---

### Note · Authors · 2024-11-15

I have read and agree with the venue's withdrawal policy on behalf of myself and my co-authors.